# Lactose Residual Content in PDO Cheeses: Novel Inclusions for Consumers with Lactose Intolerance

**DOI:** 10.3390/foods10092236

**Published:** 2021-09-21

**Authors:** Maria Sole Facioni, Simona Dominici, Francesca Marescotti, Rosanna Covucci, Isabella Taglieri, Francesca Venturi, Angela Zinnai

**Affiliations:** 1AILI—Associazione Italiana Latto-Intolleranti Aps, 55100 Lucca, Italy; presidente@associazioneaili.it; 2ELLEFREE S.r.l., Polo Tecnologico Lucchese, 55100 Lucca, Italy; sdominici@ellefree.com (S.D.); qualita@ellefree.com (F.M.); 3Department of Agriculture, Food and Environment, University of Pisa, Via del Borghetto 80, 56124 Pisa, Italy; r.covucci@studenti.unipi.it (R.C.); francesca.venturi@unipi.it (F.V.); angela.zinnai@unipi.it (A.Z.); 4Interdepartmental Research Center “Nutraceuticals and Food for Health”, University of Pisa, Via del Borghetto 80, 56124 Pisa, Italy

**Keywords:** lactose-intolerance, lactose-free, lactose cheese content, PDO cheeses, lactose-free labelling, naturally lactose-free

## Abstract

Lactose intolerance (LI) is the symptomatic condition that characterizes subjects unable to digest lactose. The main solution consists of reducing or eliminating lactose from one’s diet, and so dairy products, particularly cheeses, are often the first foods excluded. The purpose of this study is to contribute to this topic by creating an updated list of naturally lactose-free (NLF) cheeses. Twenty-five PDO (Protected Designation of Origin) cheeses were selected and analyzed to determine their lactose content. At the same time, interviews with the PDO quality control consortia were carried out to understand which parameters are involved in lactose reduction, based on the cheeses’ product specifications. The analytical techniques used here for lactose determination are the most sensitive (HPAEC-PAD and LC/MS-MS), given their low limit of quantification (LOQ) of less than 10 mg/kg. The majority of selected PDO cheeses resulted in a lactose content less than the LOQ. Because of the high variability allowed in PDO cheeses’ operative conditions, it would be better to case-by-case examine the PDO cheese specification and declare the product as NLF after repeated analysis. The results of the chemical determination of this research allowed to draw up a very useful list of PDO cheeses for both consumers and nutritionists that could be identified as NLF.

## 1. Introduction

Lactose intolerance (LI) is the symptomatic condition that characterizes those individuals who are unable to digest lactose into glucose and galactose, due to a partial or total deficiency of the enzyme lactase-phlorizin hydrolase (LPH). The symptomatology is mainly of gastrointestinal origin due to the fermentation of undigested lactose by the intestinal flora in the colon [1,2,3].

The percentage of lactose-intolerant people is around two-thirds of the world’s population, with a wide variation based on the geographical areas and countries, whereas, in Italy, it is estimated an overall frequency of about 50% of the Italian population [4,5].

The age of LI onset is typically 5 to 7 years, and the maximum clinical manifestations occur between 30 and 40 years. In populations with a high prevalence of primary lactase deficiency, the disorder normally appears around 2 years of age, while other populations with a lower prevalence show the first symptoms between 11 and 14 years. Because of the difficulties in investigation and then clinical diagnosis on toddlers, reports that focus on the clinical symptoms of lactase deficiency evident before 2 to 3 years of age are often susceptible to subjectivity. At the best of our knowledge, very few data are available in the literature about this specific topic and, in most cases, other causes must be investigated for a complete diagnosis [2].

To date, the number of lactose-intolerant people diagnosed by scientifically reliable tests, such as the H_2_/CH_4_ Lactose-Breath Test and genetic testing [6], is steadily increasing. In addition, the rising interest in LI condition is shown on Google Trends^TM^. Over the last decade, Google Trends^TM^ has been often used in various academic fields as a proxy for public interest, showing the popularity of a search term on Google. We suppose that the data shown in Figure 1 are indicative of an increased interest in LI compared to coeliac disease [7].

The main therapy for LI consists of the reduction or elimination of lactose from the diet until the symptoms disappear, depending on the form of LI. In addition, the use of oral lactase and/or probiotic, prebiotic, and post-biotic supplements is recommended, leading to an improvement in the manifestations and the composition of the intestinal microbiota [8].

Lactose is the main source of sugar in human milk and in that of the vast majority of mammals [3]. Some of the milk derivatives, due to their typical production process, may contain smaller amounts of lactose than the raw material. Among them, worldwide famous cheeses, and well-recognized Italian food delicacies, such as Parmigiano Reggiano PDO [9] and Grana Padano PDO [10], are to date the most analyzed and studied. Specific key factors in their natural manufacturing process may influence lactose reduction; one of the most known is the ageing phase [11].

To date, lactose powder is commonly used by the food manufacturing industries as an additive in many processed foods, ranging from bakery products to the least predictable, such as sausages, to improve their flavour and texture [3].

Given the large variety of food products in which lactose could be present, there is strong need for a universal lactose-free (LF) labelling, which today is still controversial. Regardless of Regulation (EU) No 1169/2011 that recognizes lactose as a substance causing adverse food reactions, a universal law regulating the labelling of “delactosed” products, defined as “lactose-free” or “low-lactose”, and the relative thresholds has not been defined yet, except for infants and follow-on low formulas (lactose less of 10 mg/100 Kcal) [8,12].

Improving lactose-intolerant people’s education about nutritional and food labelling information could represent a good strategy in order to simplify their grocery shopping. Constantly, consumers with LI have to check every label of foods and drinks they purchase, due to the widespread use of lactose in non-dairy products. For these reasons, following a LF diet is not extremely easy for lactose-intolerant people. In this regard, improving food labelling by the use of a recognizable symbol could be a good strategy to identify suitable products for consumers [13]. In Italy, the Italian lactose-intolerant patients’ association, AILI (Associazione Italiana Latto-Intolleranti APS), has assisted the creation of the first internationally registered certification mark for LF products, named Lfree^®^, that certifies and guarantees LF, naturally lactose-free (NFL), and milk-free products [8].

As shown by the eighth report of Osservatorio Immagino Nielsen GS1 Italy, the LF segment has registered a significant growth in sales percentage, also exceeding the gluten-free market (Table 1) [14].

In recent years, many food companies have aimed to produce the “lactose-free version” of their products, resulting in the proliferation of a variety of LF foods, especially dairy products [14]. In Italy, particular attention is focused on NLF dairy products that are obtained by their traditional production process. Among NLF, Parmigiano Reggiano PDO [15] and Grana Padano PDO [16] are the most advertised and claimed cheeses.

The LF segment is the fastest in sales growth among dairy products, as reported by Dekker et al. [17]. In fact, the LF dairy market is foreseen to have a steady growth reaching a €9 billion turnover by 2022, exceeding the traditional dairy products. In particular, the LF cheese category is expected to grow faster than all the other LF dairy categories, reaching an 8.4% compounded annual growth rate (CAGR) and a total yearly turnover of 632 million US dollars [17].

Scientific literature on this topic is scarce; indeed, the last research article on cheese composition was published in 2007 by Manzi et al. [18], while more recent papers were focused on the development of the analytical determination of residual lactose in hard cheeses [9,10,19].

Although lactose is enclosed in the Regulation (EU) No 1169/2011, to date, there is not an official analytical method for determining its residual content in low-lactose or LF products [12,20]. In 1996, the Italian Institution for Health (Istituto Superiore di Sanità) issued a method for determining the sugar content in food matrices.

The main analytical techniques currently reported in scientific literature to quantify residual lactose in dairy products are High Performance Anion-Exchange Chromatography coupled to Pulsed Amperometric Detection (HPAEC-PAD) [21] and Liquid Chromatography coupled to tandem Mass Spectrometry (LC/MS-MS).

Since HPAEC-PAD sensitivity, selectivity, and precision are high, this method allows to quantify low concentrations of lactose in lactose-reduced dairy products [22]. The selectivity of the method is given by an excellent separation resolution of lactose from the interferants, such as saccharides, which can form a residue from the processing of dairy products, for example, lactulose, allolactose, and epi-lactose [10,22].

Trani et al. in 2017 reported a comparison among the analytical techniques today available for the determination of lactose residues, highlighting the highest level of precision and repeatability for LC/MS-MS than the enzymatic assays and HPLC-RI method, even at the residue level, in dairy products. The reported LOQ of LC/MS-MS is 157 ng/mL, while it is 380 mg/L for the HPLC-RI method [20]. 

To date, data available on food composition, besides the scientific literature, are reported in food composition databases; the most referenced are the Food Composition Database for Epidemiological Studies in Italy (Banca Dati di Composizione degli Alimenti per Studi Epidemiologici in Italia—BDA [23]) and the Council for Research in Agriculture and the Analysis of Agricultural Economics (Consiglio per la Ricerca in Agricoltura e l’analisi dell’Economia Agraria—CREA) [24], where the lactose content in food is reported. These displayed data are not always immediately comprehensible, often reported as “trace”, or otherwise there is a total lack of information about the lactose residue. Moreover, incongruencies emerge among the sources, probably due to the techniques used to determine the amount of lactose in food.

Nowadays, the increasing number of lactose-intolerant consumers and, as a consequence, the growing demand for LF foods requires a large variety of certified products. In this context, the knowledge and perception of consumers with LI and nutrition professionals about the cheeses suitable for an LF diet were firstly investigated in this research work. Moreover, in order to provide the right information to lactose-intolerant people to make the best healthy cheese choice, the residual lactose content of the most common Italian and foreign PDO soft, semi-hard, and hard cheeses was determined. A long list of selected cheeses was evaluated for their possible inclusion in an LF diet in order to prevent any nutritional deficiencies due to their exclusion.

## 2. Materials and Methods

### 2.1. Questionnaires Provided to Consumers and Nutrition Professionals

In order to support our research, two different lists of questions were specially developed and administered through the Google Form platform to nutrition professionals and to lactose-intolerant consumers, to understand how deep their knowledge about NLF cheeses is, as well as their behavior regarding this topic.

Participation in the survey was anonymous; indeed, no personal information was collected, and approval by the ethics committee was not required. The respondents were asked to choose a predefined answer listed after a question. 

The first questionnaire was addressed to lactose-intolerant consumers and shared on the Facebook page of AILI, the Italian lactose-intolerant patients’ association, and on the major Italian Facebook’s groups of LI people. It was open for 10 days in September 2020 and 1424 consumers with LI filled it out.

The second questionnaire was addressed to nutrition professionals as medical dietitians (a physician nutrition specialist), nutritionists (enrolled in the order of biologists), and dietitians (enrolled in the order of dietitians).

It was sent by AILI’s e-mail to selected professionals and was open for 2 weeks in September/October 2020. It was filled out by 57 participants. 

All the questions are listed in the Appendix A.

#### 2.1.1. Consumers with LI Profiling

The questionnaire administered to LI consumers was filled out by 1424 participants of which 26 were excluded from the analysis because of the absence of LI. 

Of the 1398 participants included in the study, 1275 (91.2%) are females and 123 (8.8%) males. The mean age (±SD) of the subjects is 35 ± 12 years. The population is evenly distributed and representative of the Italian territory. A total of 53.8% (*n* = 752) of the interviewed individuals were diagnosed with LI for more than 3 years and 77.8% (*n* = 1088) used the Lactose-Breath Test and/or specific Genetic Test for their diagnosis (Table 2).

#### 2.1.2. Nutrition Professionals Profiling

The questionnaire administered to professionals of nutrition was filled out by 57 participants. More than a half were nutritionists (*n* = 39, 68.4%), have exercised their profession for between 4 and 10 years (*n* = 25, 43.9%), and had approximately 25 lactose-intolerant patients (*n* = 39, 68.4%). The responding nutritionists practiced their profession in 16 Italian regions even if the majority of them work in Tuscany (*n* = 20, 29.4%) (Table 3).

### 2.2. Cheese Selection

The cheese selection criteria were (1) presence of a Protected Designation of Origin label (PDO); and (2) their availability at the supermarket (considering the restrictions due to the COVID-19 pandemic).

A PDO label guarantees food properties by assuring the origin of the raw materials, the standardization of the production process, and the food nutritional and sensorial qualities, there because of the strong linkage between the product and its territory of origin [25].

Twenty-one Italian PDO and 4 PDO import foreign cheeses were selected for this study because of their large consumption. As a positive control, Mozzarella di Bufala Campana PDO was included in the cheese list. Products analyzed in this study were hard, semi-hard, or soft. Fresh cheeses were excluded because of their high lactose content due to their great aqueous content (whey) and short-term ageing.

The different types of the same PDO cheese, on the basis of ageing time, are shown in Table 4. At least two samples of different production batches were purchased for each selected cheese. Each type of selected cheese was collected from at least two different cheese factories and stored at 4 °C until the analysis.

### 2.3. Interview

The consortia and the major manufacturing companies of the selected cheeses were contacted in order to determine the range of the variability of each production phase and, as a consequence, their influence on lactose reduction. Particularly, questions were aimed to collect information about the treatment of the milk, starter culture composition, and conditions of specific production phases (e.g., pressing, forming, and ageing) (Table 5).

### 2.4. Analytical Determination of Residual Lactose

A total of 1 g of cheese sample was mixed with 1 mL of Carrez I solution, 1 mL of Carrez II solution, and Milli-Q water until reaching a ratio of 1:50 (solid/liquid extraction, *w*/*v*), and then sonicated at 40 °C for 15 min. The mixture was centrifuged (15 min, 3000 rpm) and the supernatant was filtered (0.45 µm) and then passed onto a Dionex OnGuard IIA, 2.5 mL cartridge (Thermo Fisher Scientific, Monza, Italy) before the analysis.

Subsequently, lactose residual quantification was determined using two different analytical techniques: HPAEC-PAD (Tentamus Agriparadigma Srl laboratory, Ravenna, Italy) and LC/MS-MS (Neotron Spa laboratory, Modena, Italy), according to the Italian Accreditation Body (ACCREDIA) [26]. The techniques used for lactose determination are the most sensitive and selective available, having an LOQ less than 10 mg/kg.

Cheese products from the same batch were analyzed by two different laboratories in order to verify the correspondence of the final lactose concentration. 

## 3. Results

### 3.1. Perception and Behavior towards NLF Cheeses

The first aim of this study is to assess the perception and behavior of professionals and consumers with LI towards NLF cheeses by two questionnaires suitably set up for each category of interviewees. Considering the relevant role of nutrition professionals to recommend specific diets, their knowledge about this topic was investigated. Most nutritionists interviewed (77.2%) highlighted the scarce clarity about the NLF cheeses claim. As a consequence, Gorgonzola PDO (19.4%) and Emmentaler PDO (0.9%) are low-recommended to their patients with LI (Figure 2a). Otherwise, a significant number of professionals (43.5%) mainly advise them to consume long-aged Grana Padano PDO and Parmigiano Reggiano PDO (over 30–36 months).

Regarding consumer behavior, the majority of lactose-intolerant people interviewed (54.3%) do not know the difference between an NLF product and a “delactosed” cheese. A good portion of consumers with LI (36.4%) usually purchase long-aged Grana Padano PDO and Parmigiano Reggiano PDO (over 30–36 months) while the percentage significantly decreases toward Gorgonzola PDO (15.9%) and Emmentaler PDO (13.6%) (Figure 2b).

Regarding the clarity of the NLF cheeses’ labels, the natural absence of lactose is rarely and sometimes perceived by consumers with LI (24.1% and 53.9%, respectively), so that only a minority receive this important information for the suitability of their diet (22%) (Figure 2c). Even almost half of the nutritionists confirmed that the labelling policy of NLF cheeses is not suitable for the consumers’ needs (49.1%) (Figure 2d).

In conclusion, data collected from these questionnaires confirmed the real need for the availability of an NLF cheese reference list, both for nutritionists and consumers, which were aligned in their answers (necessary or useful for 96.5% and 97.5% of the respective category of interviewed).

### 3.2. Cheeses Lactose Content

Lactose residual content was determined in the selected PDO cheeses, including Mozzarella di Bufala Campana PDO as a positive control. When the residual content was greater than the LOQ (10 mg/kg), further analyses were also performed in order to evaluate the influence of the ageing time on lactose reduction. Most of the selected PDO cheeses, starting from their first presence on the market, contained a residual lactose content lower than LOQ, as reported in Table 6. Only Pecorino Toscano PDO, at two different ageing times (t = 20 and 60 days), showed a significant lactose content: t = 20, a lactose amount of 336.8 ± 44.5 mg/kg (ranging from 284.1 to 392.9 mg/kg); and t = 60, a lactose amount of 28 ± 5 mg/kg (ranging from 21.3 to 33.8 mg/kg).

Mozzarella di Bufala Campana PDO used as positive control resulted in 3540 ± 1200 mg/kg (ranging from 2310 to 5165 mg/kg).

## 4. Discussion

### 4.1. Lactose-Intolerant Consumers and Nutritional Behavior

The higher the number of lactose-intolerant people diagnosed by scientifically reliable tests (77.8%) rather than by auto-diagnosis or non-validated tests, indicates that there is a great awareness of the LI condition, and the communication process is heading in the right direction, even if more clearness on the condition is still required.

Nevertheless, according to the majority of the nutritionists interviewed (77.2%), there is limited knowledge about cheeses to be included into the LF diet. As a matter of fact, a significant number of professionals (43.5%) gives their patients with LI the permission to consume only long-aged Grana Padano PDO and Parmigiano Reggiano PDO (over 30–36 months), even if recent publications [9,10] showed the absence of lactose starting from their first ageing time (9 months for Grana Padano PDO and 12 for Parmigiano Reggiano PDO). These data agree with the public opinion, both by the consumers with LI and nutritionists, concerning the attribution of an NLF label only to long-aged cheeses (Figure 2a,b). For this reason, Emmentaler PDO, Fontina PDO, and Gorgonzola PDO, which have a quite short ageing period (50–120 days), are not recommended by nutritionists and less-purchased by consumers with LI even if their lactose content is less than the limit of quantification (LOQ < 10 mg/kg), which is one hundred times below the limit issued by the Italian Health Ministry (<0.1%). In particular, it was observed that Gorgonzola PDO is not usually acquired by consumers with LI nor advised by nutritionists (Figure 2a,b), even if its consortium reported that the lactose content in Gorgonzola is well below the ministerial limit to define a cheese as “naturally lactose-free” (<0.1 g/100 g), as supported by a research study conducted in collaboration with CREA Research of Lodi [27].

In the last few years, multi-channel marketing has been used by consortia and cheese factories to highlight the absence of lactose in their products, but apparently it has not resulted in a deep change in people’s behavior towards these products. This supports the hypothesis that there is a strong need for a more efficient and widespread communication about NLF cheeses.

Food labelling could help consumers in their food choices in order to help them find the most suitable products for their needs. In this regard, it would be necessary that the label information be well-understood by both consumers and nutritionists.

The availability of an updated list of NLF cheeses could represent one of the main results of this research project and it could be very useful also for the cheese factories that quite often produce these kinds of products without even realizing it. Improving the knowledge on LF dairy products could allow the inclusion of NLF cheeses in lactose-intolerant people’s diet, preventing potential calcium and vitamin D deficiencies due to dairy products exclusion.

### 4.2. Cheese Analysis

In order to discuss the analytical results of the cheeses’ lactose residual content, the products were categorized into four different groups. The first one includes the most known Italian hard cheeses, such as Grana Padano PDO, Parmigiano Reggiano PDO, and Pecorino Romano PDO, that have already been studied individually. The experimental results of our research (LOQ < 10 mg/kg) confirmed the data reported in previous papers [9,10,19].

The second group of products includes cheeses (Asiago PDO, Gorgonzola PDO, Emmentaler PDO, Le Gruyére PDO, Pecorino Toscano PDO, Piave PDO, Stelvio PDO, and Montasio PDO) whose information about their lactose content are reported on their consortium’s website. In this case, according to the choice of cheese factories, the residual lactose content is not always reported on the label of the product. The analysis performed on these selected cheeses showed that their residual lactose is lower than the value reported on the consortium’s websites (10 mg/kg compared to 100–1000 mg/kg). 

The third group consist of cheeses (Taleggio PDO, Brie PDO, Fontina PDO, Provolone Valpadana PDO, Bra PDO, Caciocavallo Silano PDO, Fiore Sardo PDO, Pecorino Sardo PDO, Pecorino Siciliano PDO, Toma Piemontese PDO, and Cheddar PDO) whose information on the residual lactose content is not well-defined, or reported as traces but not specifying the value [18,23]. Considering the soft or semi-hard nature of some of these cheeses’ texture and the scarce data available, the experimental results are lower than was expected. In fact, the residual lactose content of these cheese samples is not only lower than the value reported by a previous paper [18] and/or a food composition database [23], but also less than the LOQ (<10 mg/kg).

The last group includes the cheeses whose lactose residue is completely unknown (Bitto PDO, Castelmagno PDO, and Valtellina Casera PDO) and never have been determined before this study. The residual lactose content of these selected samples was less than the LOQ (<10 mg/kg), a result that never has been shown in the scientific literature before. 

Considering all the unexpected results, especially regarding the short-aged cheeses, the role of ageing and its influence on lactose reduction in the cheese production process were investigated. In order to possibly define an NLF cheese based on its ageing time, a comparison between two different PDO cheeses having the same ageing time was performed. Pecorino Toscano PDO, a well-recognized, excellent Tuscan cheese, with a relevant role both in the national and international food market, and Asiago Pressato PDO, one of the most famous Italian cheeses from Veneto-Trentino area, were considered. The minimum ageing period of these selected cheeses was 20 days and their residual lactose content determined at this stage was not equivalent. In particular, Asiago Pressato PDO resulted in a lactose content less than the LOQ (<10 mg/kg), while Pecorino Toscano PDO, at the same stage of ageing, showed a value of 336.8 mg/kg, higher than the LOQ. 

Pecorino Toscano PDO was then also sampled at longer times of ageing: 60 and 120 days. At t = 60 days, the lactose content was 28 mg/kg, while the lactose residual at t = 120 days was less than the LOQ (<10 mg/kg). The results showed a decreasing trend in lactose content in time, confirming the role of ageing on lactose reduction, particularly within the same kind of PDO cheese. Considering different PDO cheeses, on the contrary, a univocal ageing time threshold is difficult to define, as shown from the comparison of Asiago PDO and Pecorino Toscano PDO at the same stage of ageing. Lactose residue decreases in various steps during key production phases [28]; for this reason, the product specification of these selected cheeses was studied [29,30]. The two production processes reported in the product specifications and by the consortia interviewed are similar. The processes differ mainly in the composition of their starter cultures and utilization parameters. The microbial strains composing the starter culture of the PDO cheeses represent a linkage between the product and the area from which it comes, and thus its uniqueness. In particular, the starter cultures allowed for the manufacturing of Pecorino Toscano PDO have been selected from raw and pasteurized milk of the PDO area and preserved in the official Pecorino Toscano strain collection. The species of this collection mainly used for the production of Pecorino Toscano PDO are *Streptococcus thermophilus* and *Lactococcus lactis*. On the other hand, Asiago Pressato PDO can be produced by the use of milk from the PDO area, rich in lactic acid bacteria, as a starter culture, or by the use of selected strains of *Lactobacillus bulgaricus* and *Streptococcus thermophilus*.

In addition, the metabolism of the microorganisms that compose the starter cultures and, therefore, their ability to break lactose down and utilize it as a source of energy, is deeply influenced by many parameters, such as temperature, humidity, and the pH of the various stages of the cheese production process [31].

Thanks to the information collected by the interviews of the consortia and cheese factories, the main parameters and phases potentially involved in lactose reduction observed were evaluated. In this context, the factors involved in lactose reduction were divided into primary factors, which have a great influence on the process; secondary factors, which are less relevant; and synergic factors, which help and enhance lactose loss during the whole production process. Considering that the type of milk used for conventional cheese production along with the milk thermic treatment do not affect the lactose content [3], we excluded these factors from the study. 

The exact composition used for the fermentation of all the cheeses studied was not disclosed. However, the starter culture plays a crucial role for lactose breakdown during the first steps of cheese production, as to be defined as the primary factor. Thermophilic cultures along with mesophilic cultures are the starters mainly used to determine massive lactose fermentation. Thermophilic microorganisms prevail in the first stages of lactose breakdown and *Streptococcus thermophilus* and *Lactobacillus delbrueckii* ssp. *bulgaricus* are the most representative species. Mesophilic microorganisms enhance their action on lactose reduction when the temperature decreases, and *Lactococcus lactis* ssp. *lactis* and ssp. *cremoris* are the most common species used in the fermentation process. After a period of coexistence, they gradually replace the thermophiles and become prevalent until the end of ripening. Both of these groups break down lactose into its components, glucose and galactose, using them as a source of energy during the whole production process [28,32,33,34]. 

The secondary factors are mainly of mechanical nature, allowing the physical separation of whey, in which lactose is mainly found, from the curd. The whey draining happens in all production phases, starting from the extraction of the curd from the tank to its molding, pressing, and turning. All of these latest cheese production phases alone do not have a fundamental role in lactose reduction but together they represent an essential additional factor for its physical removal [28,31].

Temperature, pH, and humidity are pointed out as synergic parameters for lactose reduction and, thus, considered relevant in each phase of the cheese production process. These factors need to be kept under strict control in order to make the process work correctly. Particularly, in the first phases of cheese production, a high humidity results in a quick lactose fermentation. During the ageing process, humidity gradually decreases, causing a slowing down of the microorganisms’ metabolism, and therefore a decline in lactose breakdown. This was confirmed by the consortia and cheese companies interviewed, as well as by a recent study [28].

Although the role of ageing in lactose reduction is well-recognized, especially in long-aged cheeses, this study contributes to redefining its role in short-aged cheeses. 

As reported by the Parmigiano Reggiano PDO’s consortium [15] and Grana Padano PDO’s consortium [16], lactose reduction is indeed substantial during the first hours of the production, declaring it as NLF at the first ageing period available on the market, which is 12 months for Parmigiano Reggiano PDO and 9 months for Grana Padano PDO. Our results are in accordance with these outcomes shown in [10]. However, as reported in [35], Parmigiano Reggiano PDO could be defined as NLF after the first 48 h from production, already having a lactose content around 0.004%. Supporting this, from the present research work it emerges that selected cheeses with an ageing period of around 20–35 days resulted with a residual lactose content less than 10 mg/kg, such as Asiago Pressato PDO, Caciocavallo Silano PDO, and Taleggio PDO.

Therefore, during ageing, the remaining quantities of lactose are gradually lost by a residual fermentation activity, resulting in a maximization of lactose reduction in the final product. The ageing phase primarily affects the texture and taste rather than lactose reduction [28,36].

## 5. Conclusions

As the LF dairy market is growing along with the attention toward health-related needs, consumers with LI and nutritionists need an improvement in LF food labelling, especially for cheeses suitable for lactose-intolerant people.

Given the lack of an official analytical method for lactose determination as a substance causing adverse food reactions, the most referenced techniques were used in this study in order to determine if the cheeses’ lactose content is less than the limit of quantification (LOQ < 10 mg/kg). The difficulties in finding a unique analytical method reliable for lactose quantification in foods are mainly due to the extremely large number of different matrices together with the high variability in their chemical/physical features. Furthermore, instrumentation costs and maintenance, duration of the analysis, and availability of specific qualified personnel represent further limitations in the choice of the analytical method. Further studies should therefore be carried out on this topic.

With the aim to create an updated list of NFL cheeses, 25 different types of PDO soft, semi-hard, and hard cheeses were collected and analyzed. All of them resulted in a minimum lactose residual content (<LOQ), except for Pecorino Toscano PDO 20 days and 60 days aged, assessing their suitability for a LF diet. Moreover, the key factors responsible for lactose reduction were taken into account, emphasizing the role of the composition of the starter culture rather than the ageing time. Even if our results identify these selected PDO cheeses as NFL, it is necessary to consider that their product specifications allow variations in the production process, which could affect the final lactose content. In addition, variability in the environmental conditions of the production area could also influence the quality parameters of the raw material. As a consequence, it is suggested to case-by-case examine the PDO cheese product specification and perform on-point analytical validations before declaring it as NLF. 

In conclusion, this research contributes to identifying the variety of NLF PDO cheeses to possibly include in a LF diet. Moreover, in order to meet consumers’ needs, it highlights the need of food labelling improvement to guide consumers in their food choices. In this regard, the Lfree^®^ certification trademark, already in use by several Italian cheese companies, could be a useful tool in order to guarantee and identify suitable products for consumers with LI. Therefore, it could be used for the immediate and safe identification of NLF cheeses, as well as provide a means to enrich the diet of lactose-intolerant people without any risk to consumer health.

## Figures and Tables

**Figure 1 foods-10-02236-f001:**
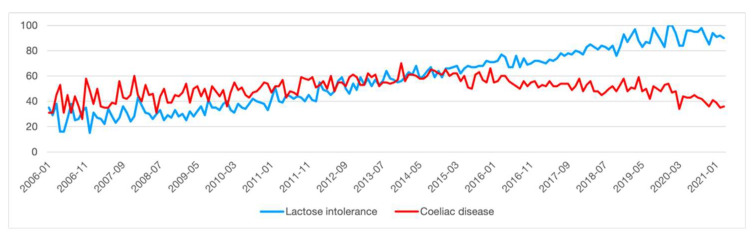
The increasing interest in lactose intolerance in recent years, from 2015 especially, compared to coeliac disease interest. The x axis indicates a period ranging from 2006 to 2021, while the y axis represents the interest normalized by Google Trends to the time and location of the query (modified by Google Trends [6]).

**Figure 2 foods-10-02236-f002:**
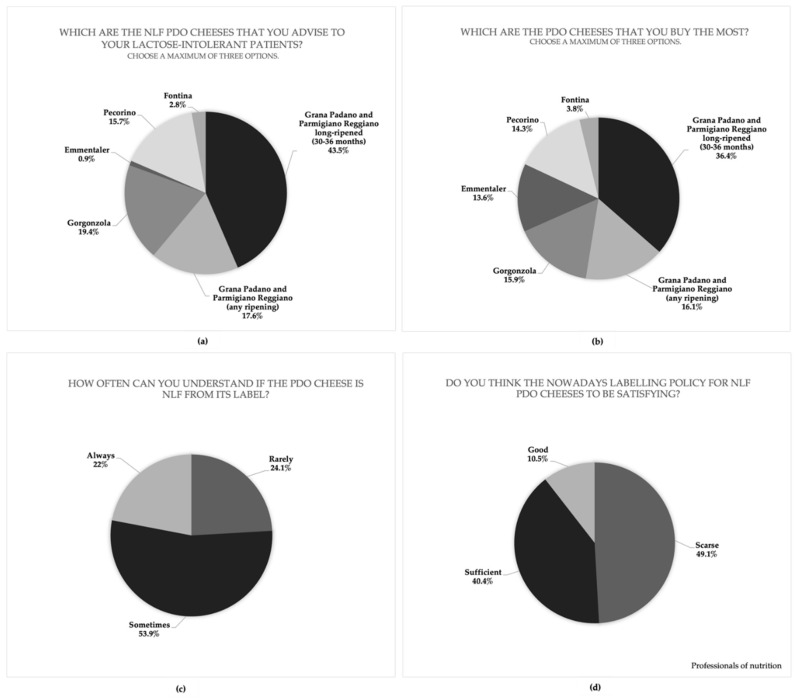
Outcomes of the questionnaires administered to professionals of nutrition and consumers with LI. (**a**) PDO cheeses recommended by professionals of nutrition to their patients with LI. (**b**) PDO cheeses mainly purchased by consumers with LI. (**c**) Understanding of the PDO cheeses’ labels by consumers with LI. (**d**) Satisfaction of the PDO cheeses’ labelling policy by professionals of nutrition.

**Table 1 foods-10-02236-t001:** Percentage growth in sales (%) of LF products compared to gluten-free products between 2015 and 2020 (modified from Osservatorio Immagino Nielsen GS1 Italy [14]).

Label	Growth in Sales (%) (2016 vs. 2015)	Growth in Sales (%) (2017 vs. 2016)	Growth in Sales (%) (2018 vs. 2017)	Growth in Sales (%) (2019 vs. 2018)	Growth in Sales (%) (2020 vs. 2019)
**Gluten free (Claim)**	0.2	4.1	1.3	1.7	4.1
**Gluten free (Logo)**	5.7	0.8	0.8	2.7	2.7
**Lactose free**	13.8	8.1	2.9	3.6	7.8

**Table 2 foods-10-02236-t002:** Sociodemographic characteristics of lactose-intolerant respondents.

Questionnaire Administered to Consumers		*n* = 1424
Question	Answers	
Are you lactose-intolerant?	Yes	1340 (94.1%)
	I am a parent or guardian of a lactose-intolerant minor	58 (4%)
	No	26 (1.8%)
Lactose-intolerant consumers		*n* = 1398
How did you find out you are lactose-intolerant?	Diagnosed by breath test and/or specific genetic test	1088 (77.8%)
	Auto-diagnosis	228 (16.3%)
	Not-validated tests (e.g., Vega-Test, Cito-Test)	82 (5.9%)
How long have you known you are lactose-intolerant?	Since birth	49 (3.5%)
	For less than 1 year	210 (15%)
	From 1 to 3 years	387 (27.7%)
	For more than 3 years	752 (53.8%)
What is your gender?	Male	123 (8.8%)
	Female	1275 (92.1%)
What is your age?	Less than 18 years	58 (4.1%)
	18–24 years	157 (11.2%)
	25–34 years	494 (35.3%)
	35–44 years	411 (29.4%)
	45–54 years	190 (14%)
	More than 55 years	76 (5.4%)
What is the Italian region you live in?	Abruzzo	37 (2.6%)
	Basilicata	14 (1%)
	Calabria	30 (2.1%)
	Campania	117 (8.4%)
	Emilia-Romagna	69 (4.9%)
	Friuli-Venezia Giulia	33 (2.4%)
	Lazio	140 (10%)
	Liguria	20 (1.4%)
	Lombardia	247 (17.7%)
	Marche	28 (2%)
	Molise	3 (0.2%)
	Piemonte	75 (5.4%)
	Puglia	131 (9.4%)
	Sardegna	111 (7.9%)
	Sicilia	93 (6.7%)
	Toscana	131 (9.4%)
	Trentino-Alto Adige	15 (1.1%)
	Umbria	15 (1.1%)
	Valle d’Aosta	2 (0.1%)
	Veneto	87 (6.2%)
What is your occupation?	Employee	954 (68.2%)
	Student	193 (13.8%)
	Non-resident student	42 (3%)
	Stay-at-home	191 (13.7%)
	Retired	18 (1.3%)

**Table 3 foods-10-02236-t003:** Sociodemographic characteristics of the professionals of nutrition interviewed.

Questionnaire Administered to Professionals of Nutrition		*n* = 57
Question	Answers	
What is your occupation?	Medical dietitians (physician nutrition specialist)	15 (26.3%)
	Nutritionists (enrolled in the order of biologists)	39 (68.4%)
	Dietitians (enrolled in the order of dietitians)	3 (5.3%)
How long have you been exercising your professional activity?	0–3 years	22 (38.6%)
	4–10 years	25 (43.9%)
	More than 10 years	10 (17.5%)
Which are the Italian regions you work in? *	Abruzzo	1 (1.4%)
	Basilicata	2 (3%)
	Calabria	1 (1.4%)
	Campania	2 (3%)
	Emilia Romagna	6 (8.8%)
	Friuli-Venezia Giulia	2 (3%)
	Lazio	7 (10.3%)
	Liguria	2 (3%)
	Lombardia	9 (13.2%)
	Marche	1 (1.4%)
	Molise	0 (0%)
	Piemonte	4 (5.9%)
	Puglia	3 (4.4%)
	Sardegna	3 (4.4%)
	Sicilia	0 (0%)
	Toscana	20 (29.4%)
	Trentino-Alto Adige	0 (0%)
	Umbria	2 (3%)
	Valle d’Aosta	0 (0%)
	Veneto	3 (4.4%)
How many are your patients in total?Indicate the numer in the year 2019.	Less than 50	16 (28%)
	51–100	14 (24.6%)
	101–200	13 (22.8%)
	More than 200	14 (24.6%)
How many are your lactose-intolerant patients? Indicate the numer in the year 2019.	Less than 25	39 (68.4%)
	26–50	12 (21.1%)
	51–100	4 (7%)
	More than 100	2 (3.5%)

* Some of the professionals of nutrition work in more than one Italian region.

**Table 4 foods-10-02236-t004:** Selected PDO cheeses and their types, classified by their ageing time, firmness, and production area.

PDO Cheese	Types	Minimum Mandatory Aging Time	Firmness	Production Area
Asiago	Pressato	Min. 20 day	Semi-hard/hard	Trentino-Alto Adige, Veneto
	D’allevo	Min 4–6 months- Max >15 months		
Bitto	n.a.	Min. 70 days to 10 years	Semi-hard/hard	Lombardy
Bra	Tenero	Min. 45 days	Semi-hard	Piedmont
Brie	n.a.	1–3 months	Soft	Est Paris area
Caciocavallo Silano		Min. 30 days	Stretched-curd hard	Basilicata, Calabria Campania, Molise, Puglia
Castelmagno		Min. 60 days	Semi-hard	Piedmont
Cheddar		Min. 9 months	Hard	County of Dorset Somerset, Devon Cornwall
Emmentaler	Classic	Min. 120 days	Hard	Bern, Switzerland
Fiore Sardo		Min. 105 days	Hard	Sardinia
Fontina		Min. 80 days	Semi-hard	Valle D’Aosta
Gorgonzola	Dolce	Min. 50 days–Max. 150 days	Soft or semi-hard	Lombardy, Piedmont
	Piccante	Min. 80 days–Max. 270 days		
Grana Padano	n.a.	Min. 9 months to >24 months	Hard	Emilia-Romagna, Lombardy, Piedmont, Trentino Alto-Adige, Veneto
Le Gruyère	D’alpage	Min. 5 months	Hard	French, Switzerland
Montasio	Fresco o Dolce	Min. 60 days–Max. 120 days	Semi-hard	Friuli-Venezia Giulia, Veneto
Parmigiano Reggiano	n.a.	Min. 12 months to >24 months	Hard	Emilia-Romagna, Lombardy
Pecorino Romano	Da tavola	Min. 5 months	Hard	Lazio, Sardinia
Pecorino Sardo	Maturo	Min. 60 days	Hard	Sardinia
Pecorino Siciliano	Semistagionato	45–90 days	Hard	Sicily
Pecorino Toscano	Fresco	Min. 20 days	Semi-hard Hard	Tuscany, Lazio, Umbria
	Semi-stagionato	Min. 60 days		
	Stagionato	Min. 120 days– Max 12 months		
Piave	Fresco	Min. 20 days–Max 60 days	Semi-hard	Veneto
Provolone Valpadana	Dolce	<60–90 days	Semi-hard	Emilia-Romagna, Lombardy, Trentino-Alto Adige, Veneto
Stelvio		Min 60 days	Semi-hard	Trentino-Alto Adige
Taleggio		Min 35 days	Soft	Lombardy, Piedmont, Veneto
Toma Piemontese	n.a.	Min 20–45 days	Semi-hard	Piedmont
Valtellina Casera		Min 70 days	Semi-hard or hard	Lombardy
Mozzarella di Bufala Campana			Fresh	Campania, Lazio, Molise, Puglia

n.a. = data not available on the label.

**Table 5 foods-10-02236-t005:** Questions administered to the consortia of the selected cheeses.

Questions Administered to Consortia
1. In your opinion, what are the phases of your cheese’s production process that determine the lactose reduction?
2. Do you think the use of raw, thermised or pasteurized milk can influence lactose content?
3. Do you think the cooking, the breaking, the extraction, and the pressing of the curd can affect the reduction of lactose?
4. Do you think the draining of the curd can influence the lactose reduction?
5. Do you think the humidity content of the product can have an influence on the residual lactose content?
6. Have you ever performed any analysis for the quantification of lactose in your cheese?
7. Other specific questions on specific steps of the production process arose after the reading of the cheeses’ product specifications.

**Table 6 foods-10-02236-t006:** Type of selected PDO cheeses and the results of the analyzed residual lactose content. The results were in agreement between the two analytical techniques used: HPAEC-PAD (Tentamus Agriparadigma Srl laboratory, Ravenna, Italy) and LC/MS-MS (Neotron Spa laboratory, Modena, Italy).

PDO Cheese and Variants Selected	^1^ Total Number of Samples	^2^ MeanLactose Content (mg/kg)
Asiago Pressato	4	<LOQ
Asiago D’allevo	4	<LOQ
Bitto	6	<LOQ
Bra Tenero	6	<LOQ
Brie	6	<LOQ
Caciocavallo Silano	6	<LOQ
Castelmagno	6	<LOQ
Cheddar	4	<LOQ
Emmentaler Classic	4	<LOQ
Fiore Sardo	4	<LOQ
Fontina	6	<LOQ
Gorgonzola Dolce	4	<LOQ
Gorgonzola Piccante	4	<LOQ
Grana Padano (9 months)	4	<LOQ
Le Gruyére D’alpage	4	<LOQ
Montasio Fresco	4	<LOQ
Parmigiano Reggiano (12 months)	4	<LOQ
Pecorino Romano	4	<LOQ
Pecorino Sardo Maturo	4	<LOQ
Pecorino Siciliano	6	<LOQ
Pecorino Toscano (20 days)	4	336.8 ± 44.5
Pecorino Toscano (60 days)	4	28 ± 5
Pecorino Toscano (4 months)	4	<LOQ
Piave Fresco	6	<LOQ
Provolone Valpadana Dolce	4	<LOQ
Stelvio	4	<LOQ
Taleggio	6	<LOQ
Toma Piemontese	6	<LOQ
Valtellina Casera	6	<LOQ
Positive control		
Mozzarella di Bufala Campana	4	3540 ± 1200

^1^ Each type of PDO cheese was purchased at least in two different production batches, in duplicate. ^2^ <LOQ: residual lactose content less than 10 mg/kg; data > LOQ are expressed as the mean ± SD of the results obtained from the two laboratories.

## Data Availability

Data available on request from the authors.

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
