# Peer review of "Lactose Residual Content in PDO Cheeses: Novel Inclusions for Consumers with Lactose Intolerance"

_foods, 2021, doi:10.3390/foods10092236_

Round 1
Reviewer 1 Report
The purpose of this study is to inform the lactose intolerant consumers about the types of cheese that are naturally free from lactose. The authors have highlighted the problem that dieticians only have very few choices of lactose free cheese to recommend to their patients. This is because the current food labeling requirement in Italy is not very clear when disclosing whether the cheese is naturally free of lactose. The study has proven that there are in fact a lot of variety of cheeses that have lactose level well below the 10mg/kg LOQ. The significance of the findings mean that it is safe for lactose intolerant consumers to consumer a wider range of cheeses that were once limited.
There are however, some concerns regarding the manuscript.
There's a lot of abbreviations used in the manuscript that were not defined properly. In the abstract, please define PDO in line 16 and LOQ in line 22.
In Figure 1, what is the units on the y axis? And how did you define/quantify "interest" in LI for Figure 1? This is not clear.
Line 66, delete "the" in front of "strong".
Line 66, define LF.
Line 70, you mean 10mg/kg not 10mg/kcal.
Figure 2, is supposed to be Table 1 not Figure 2. Please re-visit the information presented in this table. Sales variation is a wrong term to use here. It should be percentage growth in sales.
Line 119, please include the year for the Trani et al. reference,
Line 121-122, what is the reported residue level? Please quantify here.
Line 153, please define AILI?
Table 5, you have used two methods to detect the lactose level in the cheese, HPAEC-PAD and LC-MS. It is not clear if your presented results are from one of these analysis or from both. I recommend you present results from both analysis.
Line 274, "is not a suitable" should be re-worded to "is limited".
Line 402, that the role.
Line 402 - 403, the authors should explain why ageing can further reduce the level of lactose.
Author Response
Thank you so much for the time you dedicated to the revision of our manuscript. The changes you suggested have been reported in red or highlighted in yellow (when more than one Reviewer ask for the same thing) in the text.
Please see the attachment for a point-by-point response to the comments.

Reviewer 2 Report
Important study for LI consumers and health-care practitioners advising them. Provides information about a number of PDO cheese and their lactose content (or lack thereof if below 10mg/kg).
See file attached with suggestions.

Author Response
Many thanks for the time you spent to correct our manuscript. Your recommendations have been followed and the corrections are written in green or highlighted in yellow marker (when more than one Reviewer ask for the same thing), and they certainly improved the quality of the text.
Please see the attachment for a point-by-point response to the comments.
